# Survey data and human computation for improved flu tracking

Stefan Wojcik [1,6 ✉], Avleen S. Bijral[2,6], Richard Johnston[2], Juan M. Lavista Ferres [2], Gary King [3], Ryan Kennedy[4], Alessandro Vespignani[5] & David Lazer[3,5]

While digital trace data from sources like search engines hold enormous potential for tracking and understanding human behavior, these streams of data lack information about the actual experiences of those individuals generating the data. Moreover, most current methods ignore or under-utilize human processing capabilities that allow humans to solve problems not yet solvable by computers (human computation). We demonstrate how behavioral research, linking digital and real-world behavior, along with human computation, can be utilized to improve the performance of studies using digital data streams. This study looks at the use of search data to track prevalence of Influenza-Like Illness (ILI). We build a behavioral model of flu search based on survey data linked to users' online browsing data. We then utilize human computation for classifying search strings. Leveraging these resources, we construct a tracking model of ILI prevalence that outperforms strong historical benchmarks using only a limited stream of search data and lends itself to tracking ILI in smaller geographic units. While this paper only addresses searches related to ILI, the method we describe has potential for tracking a broad set of phenomena in near real-time.

[1] Twitter, 1355 Market, St. San Francisco, CA, USA. [2] Microsoft, One Microsoft Way, Redmond, WA, USA. [3] Harvard University, Cambridge, MA, USA. [4] University of Houston, Philip Guthrie Hoffman Hall, Houston, TX, USA. [5] Northeastern University, 177 Huntington Ave, Boston, MA, USA. [6]These authors contributed equally: Stefan Wojcik, Avleen S. Bijral. ✉email: swojcik@twitter.com

The COVID-19 pandemic has highlighted the need for fast and actionable forecasts of health threats. Mining search and social media data for real-time tracking (i.e., "nowcasting") of flu prevalence and other events has become a major focus in public health, computer science, and other disciplines[1–10]. One highly cited example of using search query data to forecast illness is Google Flu Trends (GFT). Many efforts have built upon the early work of GFT, which provided a promising real-time influenza tracking system, based on the idea that searches for the flu will increase when users are ill. While GFT showed the value of using digital streams like search data to forecast important events, it ultimately faced considerable challenges[1,11,12]. We pick up on two major issues identified by prior research[12], and present methods, relying on standard techniques, to address them.

The central premise underlying the GFT methodology—and one which still underlies many efforts of its kind today—was to use the massive amount of search queries produced by users to find a few that are the best at predicting (Centers for Disease Control and Prevention) CDC flu rates. The first issue researchers pointed out was that GFT was a problematic marriage of big and small data, since the algorithm involved over 50 million search terms narrowed down to predict a little over 1000 CDC data points[12]. This methodology doomed GFT to sweep up some false positive search terms that happened to peak at the right time and place. Rooting out false positive predictors in such a massive predictor space is a central concern in the use of search data and other big data streams, but, as a consequence, the GFT model missed its CDC target for long stretches of time[12]. Later work corrected some of the errors of the GFT approach, by incorporating reports from CDC as the season progresses, adjusting for changes in search behavior, and leveraging the properties of time-series, to vastly improve the reliability of the forecast[9]. But the central approach—mapping a huge number of search queries to a short series of CDC data—remains largely the same.

The second major issue identified by prior researchers was measurement of flu-like illness itself. For systems like GFT, search query data are the primary instrument for measuring influenza-like illness (ILI) in the population. However, it is not yet clear that search query data are even a good indicator for ILI symptoms[12]. Sparse empirical evidence currently exists to support (or reject) the idea that users experiencing ILI symptoms make more searches for the flu. To generate such evidence, one would need access to a user's online searches and information about any illness symptoms they are currently experiencing. Such data would allow researchers to observe whether the presence of illness symptoms associated with the flu leads to an increased propensity to search for flu-related information. However, even with high-quality observational data from a user's online searches and reported symptoms, more information is needed. The prior distribution of search propensity is unlikely to be uniform across all demographic groups, and is likely to be correlated with ILI prevalence within groups. In addition, users experiencing ILI symptoms may make searches to find diagnostic information, but only if they cannot access that information through other means.

In an effort to address these two primary areas of concern, we describe an approach to flu tracking that uses standard social-scientific methods to build a behavioral model of the relationship between Internet search and flu-like illness. Our approach captures user search data combined with survey data of flu-like symptoms to give us a granular view of the measurement properties of search data. The behavioral model based on these data demonstrates the underlying measurement value of using search data as a surrogate for flu-like symptom reports at the household level. We enlist trained coders to remove false positive predictors of flu-like experience that are otherwise challenging to do algorithmically. We leverage our behavioral model to create a weighted forecast of ILI in the US population. We find that this survey-driven approach enables us to define the demographic differences in search, detect relevant flu searches in a more systematic fashion, and adjust for demographic variance in flu search behavior when building a forecasting model. This method could be useful in tracking various other phenomena for which survey data might provide an empirical link between large-scale digital sources (such as queries or social media posts) and local/regional outcomes (such as ILI, unemployment rates, or movie ticket sales).

In comparison to existing approaches, our model can track ILI prevalence in subregional populations up to 2 weeks in advance of CDC flu reports. To demonstrate the practical application of our model to a digital search stream, we compare its performance against strong baseline models, including a highly accurate auto-regressive LASSO model of the historical signal[9]. There is a long history of epidemiological models that estimate rates of transmission through populations using a mix of network data, social media data, and clinical data[1,2,4–6,8,9,13,14]. Statistical models use sample data to infer population-level rates of ILI, while mechanistic models describe whole populations to simulate ILI spreading and infer rates of ILI infection[13]. Both types of modeling approaches estimate rates of ILI in different populations in order to generate advanced warnings of season start time, peaks, or duration, often to substitute for costly and time-consuming clinical measures (like the CDC flu reports). However, common metrics for model evaluation remain a topic of some debate[13]. We focus here on applying our survey results to build a tracking model that uses search queries effectively to track ILI prevalence.

## Results

**Behavioral and tracking models.** Our approach is to directly measure how ILI symptoms affect searches associated with the flu. To do so, we tracked the online behavior of a set of users for an entire flu season, and identified user searches and online behaviors pertaining to ILI diagnostic information. We will discuss two models: a behavioral model based on a case–control design that maps Internet search behavior to ILI symptoms, and a tracking model that maps behavioral signals from search queries to generate predictions of flu prevalence. The behavioral model relies on a survey in concert with browser tracking to tabulate whether users had ILI symptoms during the flu season. Then, we build a flu tracking model drawing on insights from the behavioral model. We tested the flu tracking model using data from national and state-level CDC reports.

**Survey and browsing data.** We partnered with a survey vendor maintaining a nationally representative panel of approximately 20,000 individuals with personal computers (Supplementary Methods 1.3.1 and Supplementary Table 1). Respondents in the panel had consented to participate in marketing research in return for monetary compensation; researchers were able to track their web browsing, their search activities, and send questionnaire invites. All survey and panel data were anonymous and purged of any personally identifiable information before they were received for analysis.

Since users who conduct a higher volume of searches are more likely to generate a flu-related search by chance, we selected subsets of the ongoing panel to participate in flu forecasting research. We sent survey invites to two subsets of participants ($N = 1180$ and $N = 4000$) from the full 20,000 person panel. One set of participants met the following criteria: (1) they had executed queries in any search engine (including Bing, Google, Yahoo), (2) they had used flu-related keywords (e.g., "flu,"

"fever," "influenza," "swollen," "cough," "pneumonia," "sore throat"), or they had visited flu-related URLs (e.g., WebMD, CDC, Wikipedia). The second group was a comparison group that did not execute a flu-related query or visit a flu-related web site. We used a case–control approach to select the comparison group, drawing randomly from the panel within bins based on search volume. We did this to maintain balance on search volume since people with flagged searches had higher than average search volumes (Supplementary Methods 1.3.1 and Supplementary Table 3).

From these individuals, we collected survey responses for a total of 654 individuals (13% response rate), broadly similar to the invitees, of which 10 did not have any reported search volume in the sample period. This left us with 262 who had searched a flu-related keyword or site and 382 who did not (omitting 10 who had no reported search; Supplementary Methods 1.3.3 and Supplementary Table 1).

The panel provided access to the entire browsing history of respondents, allowing us to examine web page visits and search queries simultaneously. Incorporating information about flu-related web visits, in addition to queries, allowed us a more complete picture of user behavior in the presence of flu-like symptoms. Table 1 shows means and standard deviations of the main survey variables of interest. Volume refers to the logged number of searches by a respondent, and ILI is defined as when respondents report both fever and cough for themselves or family members.

We fielded our flu survey in the spring of 2015. Our survey questionnaire asked respondents for demographic, household, and flu-related information. We asked respondents about all symptoms of ILI since November 2014, followed by a question in which we asked which month these were experienced. We also asked these questions about children and other adults in the household (including spouses). We followed up by asking which sources (Supplementary Methods 1.3.3), if any, these individuals used when seeking information about these symptoms and health care provider diagnoses, if relevant.

**Identifying flu-related searches**. A central question surrounding the use of query/social media data for flu prediction is how to identify a flu-related query, post, or web page. The traditional approach used by GFT and others is to narrow down searches based on their observed association with flu prevalence, but this method produces a large number of false positives. Instead, we identified flu-related search queries with the aid of trained human coders. This method allowed us to select search queries with a high prior likelihood of being associated with the flu and to exclude closely related yet irrelevant queries. For example, using this method we were able to identify and remove searches associated with news and current events unlikely to reflect underlying

symptoms of users, such as "Obama sore throat," which with other methods may be misleadingly associated with flu prevalence.

Broadly, flu-related search activity can be defined as when search text contains key words and cues likely to be relevant to a person experiencing symptoms and/or seeking diagnosis information. These may include simple searches for flu-like symptoms —including fever, cough, sore throat, and other canonical symptoms—as well as more specific searches, such as "what are the symptoms of the flu?", or "continued fever is a symptom of..." We considered such queries to be highly relevant to user experience and labeled such queries as A1 type searches. Other types of queries related to research or news-related queries (such as "Spanish flu"), we labeled as A2. We also labeled other types of queries, such as those associated with secondary symptoms of ILI, non-ILI illnesses, and other categories (see Supplementary Methods 1.5.2).

To identify A1 (flu-related) searches and browsing behavior from respondent data, we asked trained human coders to label each search query and web page visited by respondents in a multi-step process (Supplementary Methods 1.5.4). We found 21% of respondents made an A1 query or page visit, and 14% made an A2 query or page visit. With these labels in hand, we modeled the relationship between reported flu-like symptoms and online activity.

**Behavioral model of search**. We identified respondents as having ILI symptoms if they reported having both fever and cough at some point during the flu season. The presence of ILI symptoms for the respondent or in the household served as our key explanatory variable linking ILI symptoms to A1 search behavior (our key outcome variable). However, in estimating our model we also included an array of demographic characteristics of the respondents to adjust for variation across demographic groups and for heterogeneous effects of ILI symptoms on search behavior.

To infer the effect of the combination of symptoms on flu-like search activity, we used a classic (or cumulative) case–control design to estimate the relative risk of A1 search when flu symptoms are reported[15]. To do so, we paired respondents who made A1 searches (positive cases) with others who did not (negative cases), and estimated the effects of ILI symptoms and demographic variables. Because A1 queries are a relatively small share of all searches made online, we adjusted our estimates for differences between the base rate of flu-like search in our sample and our best estimate of the rate of flu-like search in the population. We used the average flu-like search rate in the Bing search engine to make this adjustment (Supplementary Methods 1.6.1).

Based on this model, we calculated the relative risk (RR) and risk difference (RD) of A1 search activity given ILI symptoms: $\mathbf{RR} = Pr(Y = 1|X_1, \tau)/Pr(Y = 1|X_0, \tau)$, $\mathbf{RD} = Pr(Y = 1|X_1, \tau) - Pr(Y = 1|X_0, \tau)$[15], where $\tau$ is the incidence of A1 searches, $X_1$ is a $k$-vector of covariates of a treatment group with flu symptoms and $X_0$ indicates a $k$-vector of covariates of a control group lacking flu symptoms. Covariates included search volume, gender, parenthood status, and age (Supplementary Methods 1.3.2 and Supplementary Table 5). To control for the fact that the population's rate of flu-like (A1) search differed from our sample, we substituted the constant term in the logistic model for a corrected term that matched the observed rate of flu search in the Bing search engine. This corrected term is calculated as: $B_0 - ln[\left(\frac{1-\tau}{\tau}\right)\left(\frac{\bar{y}}{1-\bar{y}}\right)]$, where $B_0$ is the original constant term, $\tau$ is the rate of A1 search in the population (1.2e-5), and $\bar{y}$ is the rate of A1 search in the sample. The coefficients remain unbiased.

**Table 1 Descriptive statistics of survey data.**

| Statistic | N | Mean | SD | Min | Max |
|---|---|---|---|---|---|
| Volume | 644 | 5.688 | 1.610 | 0.000 | 9.492 |
| Female | 654 | 0.610 | 0.488 | 0 | 1 |
| Parent | 654 | 0.315 | 0.465 | 0 | 1 |
| Spouse | 654 | 0.509 | 0.500 | 0 | 1 |
| Age | 654 | 4.610 | 1.434 | 1 | 7 |
| Household ILI | 654 | 0.349 | 0.477 | 0 | 1 |
| Respondent ILI | 654 | 0.245 | 0.430 | 0 | 1 |

Rows are means, standard deviations, minimums, and maximums of survey variables included in behavioral model. Age is a numeric variable indicating which age group respondents belonged to, between 18 and 65+ years.

**MRP smoothing and re-weighting**. We construct smoothed and re-weighted estimates of A1 searches by day and by state using multilevel regression with post-stratification (MRP). MRP is a method for making predictions with non-representative and/or non-probability survey data[16], typically at the sub-national level. This is useful for practitioners because it allows for modeling a wider array of data than models that require nationally representative data. To apply this approach, we first assign binary A1 labels to a large corpus of Bing search queries. This comprises our response variable. Next, we estimate the proportion of A1 search queries in state $s$ within a moving time window, producing a prediction for the final day of the moving window. We re-weight each state-day prediction based on the number of zipcodes in each state belonging to each cell (stratum)(Supplementary Methods 1.8:2). MRP effectively splits the data into cells that represent unique combinations of characteristics. For example, we would identify a cell for zip codes in Minnesota with a high number of college graduates if we grouped the data by state and education.

With a relatively small sample of users, we do not observe all possible word combinations of flu-like searches. This is problematic for forecasting, because we may miss many positive indicators of flu-like experience. In order to capture a wider array of potentially flu-related queries, we used our labeled sample of queries from the respondent panel and applied an embedding method called DOC2VEC[17]. The DOC2VEC method creates document representations based on word embeddings learned from a corpus of text. These word embeddings capture deeper co-occurrence relations that allow us to retrieve similar documents. In our case, this method located other queries that are semantically related to labeled A1 queries but were not in our browser dataset. We then tasked human coders with applying the same A1 labeling scheme we described earlier to the new and expanded set of A1 candidate queries from DOC2VEC.

The MRP method leverages information from similar cells—those possessing one or more similar characteristics—to reduce variance where Internet search might be rare. This reduces the amount of error for sparse cells and is useful for datasets with sparse coverage for certain population subgroups. It re-weights estimates based on census benchmarks for the geography where one desires to make a prediction. We acquired census data from the American Community Survey 2014 5-year estimates (Supplementary Information 1.7.2). We created 3131 cells based on combinations of state, proportion of children per household (binned by quartile), proportion possessing a college education (binned by quartile), and age in each zipcode (for more information on how the variables were constructed, see the Supplementary Information 1.7.2). We deploy the following MRP model over each window of time to smooth and re-weight search queries collected at the zip code level:

$$Pr(y_{it} = 1) = \text{logit}^{-1}\Big(\beta_0 + \beta_{[it]}^{\text{Income}} + \alpha_{j[it]}^{\text{State}} + \alpha_{j[it]}^{\text{Education}} + \alpha_{j[it]}^{\text{Age}} + \alpha_{j[it]}^{\text{Child-per-House}} + \alpha_{j[it]}^{\text{Education*Age}}\Big) \quad (1)$$

Here, the subscript $j[i]$ refers to the cell ($j$) and the window ($t$) to which the $i$th query belongs. The response variable $y_{it}$ indicates whether the $i$th query in that window is labeled A1. We applied the MRP model over a rolling 3-day window of all search queries possessing a flu-related term, making a prediction for the final day of the window (Supplementary Information 1.7.2)[18]. We then re-weighted to state- and national-level census benchmarks using the formula in ($\hat{y}_s^{PS}$) below (PS = post-stratified)[19].

The term $\beta_{[it]}^{\text{Income}}$ is a fixed coefficient on income at the zipcode level, as the inclusion of predictive geographic covariates often improves the performance of MRP models[20]. The terms $\alpha_{j[i]}^{\text{State}}$,

$\alpha_{j[i]}^{\text{Education}}$, $\alpha_{j[i]}^{\text{Age}}$, etc. represent varying coefficients associated with each categorical variable. Effects are assumed to be drawn from a normal distribution and estimated variance. For example, for states it is assumed that $\alpha_j^{\text{State}} \sim N(0, \sigma_{\text{State}}^2)$.

To create state-level estimates, we re-weighted each estimate based on the number of cells of each type in each state, following[19]:

$$\hat{y}_s^{PS} = \frac{\sum_{j \in J_s} N_j \hat{y}_j}{\sum_{j \in J_s} N_j} \quad (2)$$

Once we obtained smoothed daily state-level estimates of A1 searches, we then trained different time-series models on state- and national-level CDC data for the ILI rate between 2012 and 2016, using the MRP estimates as an exogenous signal. We also trained comparison models using an unprocessed A1 exogenous term and others using historical data alone.

**Time-series forecasting models**. We evaluate the merits of the re-weighted MRP search signals by employing them as inputs in forecasting models. Our goal is not to demonstrate any new time-series algorithm but to demonstrate that already effective forecasting models can get a meaningful performance boost by using the MRP signal.

As demonstrated by Lazer et al.[12], ILI rates have a strong historical and seasonal dependence, and a model trained purely on history can be a strong predictor of future influenza rates. So we considered, based on the current literature, different models that incorporate the relevant exogenous signals derived from our survey and compared the results to these models based only on the historical ILI signal[9]. This included both the popular Lasso-based[9] as well as the SARIMA-based methods. Let $Y^{\text{ILI}}(t)$ be the time series of weekly influenza rates for a geographic entity (US or State), $X_j^{\text{A1}}(t)$ be the time series of logit transformed volumes of A1 labeled search queries and finally $X_{\text{mrp}}(t)$ be the exogenous weekly time series corresponding to the MRP signal aggregated at the national level (see Table 2). We also assumed that $\epsilon(t) \sim \mathcal{N}(0, \sigma^2)$.

The following models were then considered

$$\text{SARIMA} - \text{HIST} : \phi_p(B)\Phi_P(B^s)Y^{\text{ILI}}(t) = \theta_q(B)\Theta_Q(B^s)\epsilon(t) \quad (3)$$

$$\text{SARIMA} - \text{MRP} : \phi_p(B)\Phi_P(B^s)Y^{\text{ILI}}(t) = \theta_q(B)\Theta_Q(B^s)\epsilon(t) + \phi_1 X_{\text{mrp}}(t) \quad (4)$$

$$\text{SARIMA} - \text{A1} : \phi_p(B)\Phi_P(B^s)Y^{\text{ILI}}(t) = \theta_q(B)\Theta_Q(B^s)\epsilon(t) + \phi_1 \sum_{j=1}^{m} \frac{X_j^{\text{A1}}(t)}{m} \quad (5)$$

**Table 2 National ILI accuracy (horizon = 1 week) scores based on percent CDC ILI visits.**

| Method | RMSE | MAPE | MAE |
|---|---|---|---|
| SARIMA-HIST | 0.261 | 8.266 | 0.175 |
| SARIMA-MRP | **0. 234** | **8. 058** | **0. 157** |
| LASSO-HIST | 0.279 | 11.82 | 0.207 |
| LASSO-A1 | 0.270 | 9.342 | 0.182 |

The rows indicate a model loss—root mean squared error (RMSE), mean average percent error (MAPE), and mean average error (MAE). See Supplementary Information 6.8 for loss calculations. Bold text indicates minimum values.

**Table 3 State-level tracking model RMSE ($h = 1$ week/$h = 2$ weeks).**

| Signal | NM | DC | DE | NY |
|---|---|---|---|---|
| SARIMA-HIST | 97.70/146.93 | 32.85/53.62 | 58.26/90.21 | 643.18/1270.72 |
| SARIMA-MRP | **78.72/113.73** | **26.59/38.88** | **54.27/82.07** | **503.29/952.86** |
| SARIMA-A1 | 82.53/128.41 | 27.42/39.74 | 58.44/88.70 | 513.75/935.47 |

Bold text indicates minimum values.

$$\text{LASSO} - \text{HIST}: Y^{\text{ILI}}(t) = \sum_{i=1}^{p} \theta_i Y^{\text{ILI}}(t - i) + \epsilon(t) \qquad (6)$$

$$\text{LASSO} - \text{A1}: Y^{\text{ILI}}(t) = \sum_{i=1}^{p} \theta_i Y^{\text{ILI}}(t - i) + \sum_{j=1}^{m} \phi_j X_j^{\text{A1}}(t) + \epsilon(t) \qquad (7)$$

for the seasonal ARIMA model with exogenous variables (SARIMA-MRP/SARIMA-A1) or without (SARIMA-HIST) the notation refers to a ARIMA($p, d, q$) × ($P, D, Q$)$_s$ model in the Box–Jenkins terminology[21]. The LASSO-A1/LASSO-HIST models were estimated using a Lasso penalty. ARGO[8,9] is an example in the context of ILI prediction.

For SARIMA-based models we chose the appropriate orders ($p, P, q, Q$) for the model using AIC as a criterion. We set $p = 52$ in the LASSO-based models to account for the seasonal effect in ILI rates, but, since we were using the Lasso penalty, the coefficients of most of these lags did not appear in the final model. This approach also served to select the appropriate lag in modeling ILI rates. Finally, we provided predictions for the 2015–2017 seasons by using a rolling 3-year period to train and predict.

To examine how well our model performed at finer spatial granularity than the national level, we collected flu data on the number of positive influenza swabs from the states of DE, DC, NM, and NY (see figures in Supplementary Methods 1.7.4 and Supplementary Fig. 8). These states were selected because they (1) collect and publicly post current flu prevalence rates on their official health pages and (2) make historical flu prevalence data available for long enough time spans to be useful for prediction. However, unlike the CDC data these states do not provide a total number of hospital visits as the denominator to the ILI rate. Instead, we trained our model using the raw counts of the ILI positive cases reported in the state. These state-level search signals will be especially noisy due to lower search volumes than the national level.

We used a rolling 3-year period to train and test our models. We trained using the MRP smoothed-reweighted signal (SARIMA-MRP), and made comparisons against a history-only model (SARIMA-HIST) and an averaged simple unweighted (non-MRP) A1 series (SARIMA-A1)(see Table 3). All of these models are exceptionally strong, as they all use out-of-sample performance to parameterize the number of lags to include. To compare the different methods we used RMSE as a metric after selecting the model parameters based on the AIC (Akaike Information Criterion). We tested the performance of our model on rolling 1- and 2-week ahead forecasts using exogenous signal at the most predictive lag.

**Key survey findings.** Before discussing how online searches were associated with reported symptoms, it is worth noting that we found differences in the relative rate at which respondents said they sought information from healthcare providers versus online sources. Fully 33% of respondents said that they searched for information online about symptoms they had experienced but did not consult a health provider. 26% consulted the Internet and a health provider, and only 6% reported seeking information from a health provider and did not consult the Internet, 28% did

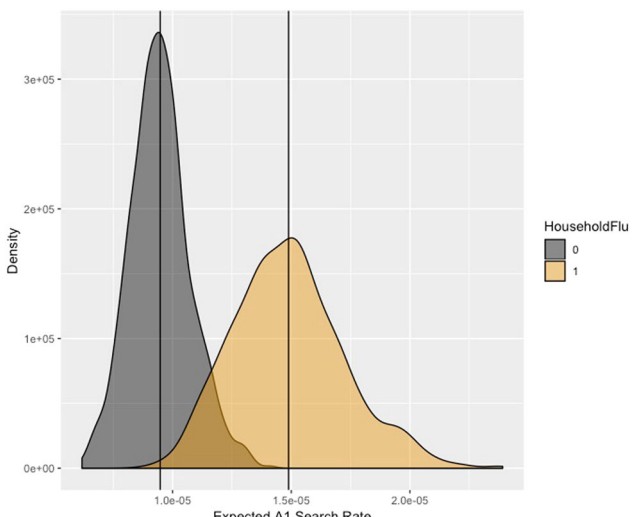

**Fig. 1 Estimated search rates by flu presence.** Estimated population-level search rates. Based on survey data in the presence of reported household flu (in light yellow) and absence of reported household flu (in dark grey). The x-axis indicates a proportion of searches and the y-axis indicates the estimated density of searches where household flu = 0, 1.

neither of these things and the remainder refused to answer (Supplementary Methods 1.4).

We observed a heightened tendency to search for flu (A1) when there was an occurrence of flu symptoms in the household. Fig. 1 plots the expected values of $Y$, $Pr(Y = 1|X, \tau)$, where household flu is present and absent. In Fig. 1, the means of $= Pr(Y = 1|x_1, \tau)$ and $Pr(Y = 1|x_0, \tau)$ are shown with dark vertical lines. The estimated risk ratio (RR) is 1.57 (95% CI = 1.05, 2.34), meaning that those with flu symptoms execute almost 60% more A1 searches compared to those exhibiting no symptoms at all. Since the base rate of query activity in the population is relatively low, this amounted to a risk difference (RD) of about 5.41e-06 (95% CI = 5.57e-07, 1.09e-05). In other words, we found the presence of flu-like symptoms increased the relative rate of flu-like search queries (A1) by more than half, but when calculated as a change in the proportion of all searches, the difference is small. This finding underscores why using individual-level data to build a behavioral model is important, detecting such minor changes using query data alone would be quite challenging. We generally did not find symptoms to positively affect A2 search queries (Supplementary Information 6.1).

Searches were noisy indicators for particular subgroups, specifically heavy searchers, women, and mothers. We found that A1 searches were correlated with higher search volumes generally, suggesting that individuals who search for a lot of information online make flu-related searches even in the absence of symptoms. A person in the third quartile of search volume was about 30% more likely to search for the flu (RR=1.32, 95% CI = 1.04, 1.68) in comparison to someone with the first quartile

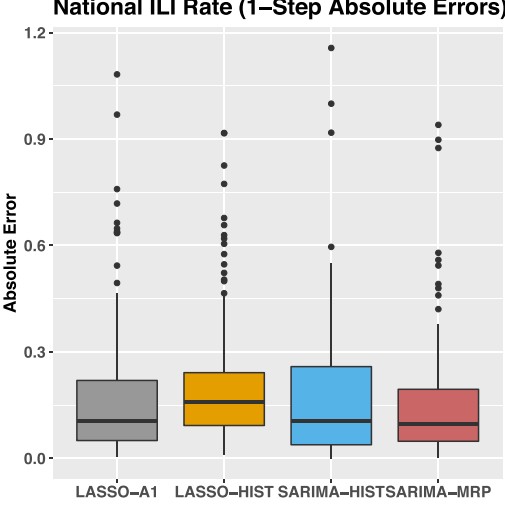

(a) Absolute Error boxplot.

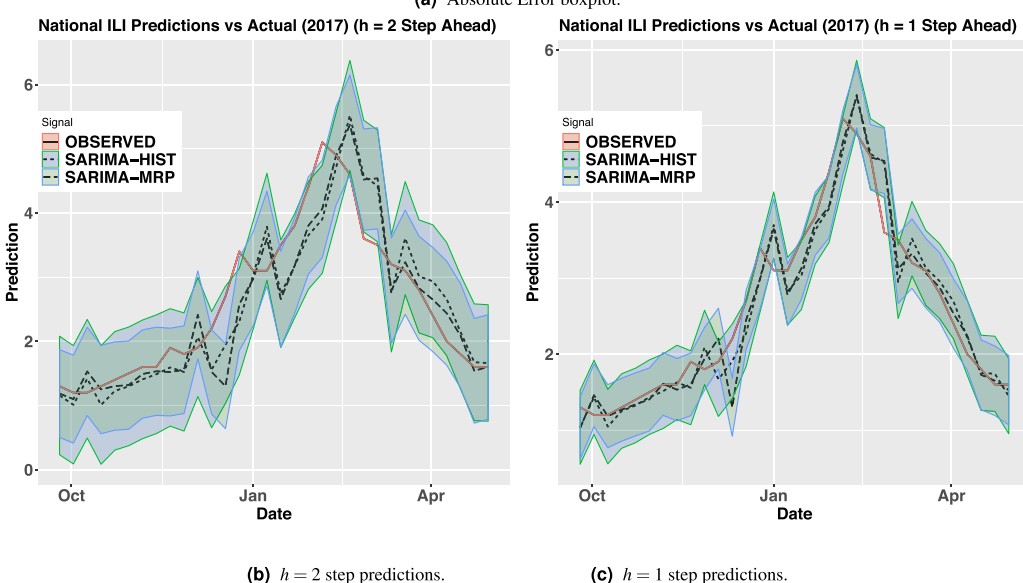

(b) $h = 2$ step predictions.　　　(c) $h = 1$ step predictions.

**Fig. 2 National tracking model performance metrics. a** Absolute error boxplot. **b** $h = 2$-step predictions. **c** $h = 1$-step predictions. **a** Box and Whisker plot of the absolute error of multiple model predictions based on the national percent of ILI visits based on CDC data ($T = 123$ weeks). Horizontal line is median, boxes indicate 25% and 75% quantiles, and whiskers extend no more than 1.5 the length of each box. **b/c** The $x$-axis depicts the 2017 date and the $y$-axis depicts the predicted/true national percent of ILI visits per CDC data. Solid line is the actual CDC ILI visits and the dotted lines are point forecasts based on the SARIMA-HIST and SARIMA-MRP models along with upper and lower prediction intervals for both $h = 1$ and $h = 2$-steps ahead. ($T = 123$ weeks).

search volume, with a risk difference of RD = 3.09e-06 (95 CI = 4.62e-07, 6.10e-06)(See Supplementary Information Table 13).

Fathers and mothers had dramatically different behaviors in reaction to perceived child illness. We found fathers to be much more likely to make flu searches when their children were ill with ILI symptoms, whereas mothers tended to have a high baseline tendency to make such searches regardless of child illness. Fathers made more than 8 times as many A1 searches when their children exhibited ILI symptoms compared to when they were symptom-free (RR = 8.75, 95% CI = 2.21, 42.36), which amounted to a risk difference of RD=1.95e-05 (95% CI = 5.16-06, 4.53e-05).

We found A1 search rates were not highly distinct across respondent race and education. Differences between self-identified racial categories were not significant. This included individuals self-identifying as White, Black, Asian, Native, or Hispanic. Similarly, education was not a strong differentiator of A1 search among respondents. Individuals who reported having completed high school were not more or less likely to make flu

searches compared to individuals who reported completing some college, a bachelor's degree, or a graduate degree.

**Tracking model performance**. Table 2 shows that the MRP signal improves upon the 1-week ahead MAPE (mean average percent error), RMSE (root mean squared error), and MAE (mean absolute error) compared to the history-based models (for a total period of $T = 123$ weeks) (all metrics look at the average deviations of model predictions, see Supplementary Information 6.8 for mathematical definitions of these metrics). As such, estimates in 2016/2017 deviated by 12.29/13.74% from the true values in these tests. For reference to a more common metric, the MRP tracking estimate correlates at 0.90/0.84 with the national ILI rate in the 2016/2017 season. Fig. 2a shows that the absolute error for the MRP signal has a lower mean and variance than other history or A1-based models. On average, the SARIMA-MRP model showed fewer outliers and a smaller median error compared to the pure history model. We conjecture that since

Bing is only a portion of all search queries, our models would perform better with more search data. Additionally, if we look at $h = 2$-week ahead predictions (the prediction at step $t + h$, given the CDC data point at step $t$ and the exogenous signal at step $t + h$), the MRP signal proves to be even more valuable achieving a RMSE/MAPE/MAE of 0.386/12.855/0.263 vs. 0.456/13.584/0.299 using only history (SARIMA). Looking at the plot for our forecasts, Fig. 2c and b show that SARIMA-MRP predictions are better aligned with the observed national ILI rate with tighter upper and lower prediction intervals for both the 1-step and 2-step ahead predictions. These 95% intervals are given by $\hat{Y}^{\text{ILI}}(t + h) \pm 1.96\sqrt{\hat{\sigma}_h}$, where $\hat{Y}^{\text{ILI}}(t + h)$ is the point forecast at time step $t + h$ and $\hat{\sigma}_h$ is the variance of the estimated noise terms (assuming Gaussian errors). More details can be found in[21].

Pure history-based models can be very powerful if appropriately trained and tested. To illustrate this, we note that the history-only SARIMA-HIST model outperforms all other approaches in tracking national CDC estimates, save for our demographically sensitive SARIMA-MRP approach. For another recent SARIMA-based Influenza model please see[22].

**State-level performance**. The SARIMA-MRP model performed better than history alone in all four states, and in all the cases it was the outright best model (based on error rates). The improvement of the SARIMA-MRP model was especially sharp with a 2-week horizon. Table 3 displays the root mean squared errors for a horizon of 1 and 2 weeks in each state.

## Discussion
In this paper, we tackled a subset of the problems of ILI prediction in the context of search query data. We offer two primary advances. First, existing work has operated without an underlying behavioral model linking search with illness. Here, we use survey data linked with user behavior to observe the connection between ILI symptoms and search behavior more directly than prior research. We find that there is substantial variation in the propensity of users to execute searches in the presence of ILI symptoms based on demographic factors. Next, we leverage these insights and integrate demographic data to account for the uneven propensity of flu search and to build an example ILI tracking model. The MRP-SARIMA tracking model outperforms models containing historical or exogenous signals at 1- and 2-week horizons.

The performance of the MRP-SARIMA model is likely influenced by a couple of factors. First, our model tracked ILI prevalence at the state and national level using demographically smoothed (noise-reduced) estimates of flu volumes. Smoothing reduces sensitivity to sparsity among certain population subgroups, reducing error[19]. The model also re-weighted (bias-reduced) estimates of flu-like query volumes, which brought estimates more in line with known underlying population values. Taken together, demographically re-weighted search signals and human-aided query labeling produces an accurate and near real-time model of flu prevalence at the national and state levels.

There are important limitations to this approach. First, research suggests that asking respondents to recall symptoms at increasing lengths of time creates measurement error (Supplementary Information 3.4)[23–26]. Health survey research suggests that extending recall windows leads to underreporting of symptoms[24,25], while research in public opinion research suggests that dating error is unbiased, but variance increases linearly as the recall period increases. Another recall-related issue is boundary effects—where asking respondents whether they experienced flu or cough from November 1, 2014, to the present, may create errors in judgment that can only be later in time than the start and earlier in time than the beginning of the interval—so errors may pile up near the center of the interval. We checked for differences between respondents who completed the survey earlier compared to those who completed it later, and found no differences (Supplementary Information 3.4).

Next, it is challenging to connect all searches to one individual, even if they have indicated they were the sole user of the device (see additional analysis in Supplementary Information 6.2). Some error may stem from searches using devices not containing the tracking software (if a respondent happened to use a friend or spouse's device).

Finally, comparison of our ILI tracking model to the state of the art is challenging, given our relatively modest survey sample, focus on subregional estimates, and relatively limited search stream. We expect that beginning with a larger survey sample and greater search coverage is likely to yield superior results. Similarly, starting with survey data limits the types of search queries that will be observed, necessitating the use of co-occurrence methods such as the DOC2VEC approach. Here, future research might identify A1 searches separate from the survey using a larger search stream. This would encompass a broader range of queries at the outset and obviate the need for DOC2VEC expansion.

We tested the practical value of our approach by defining a model to track state-level ILI rates and national ILI rates. The results suggest that constructing a behavioral model of search can yield practical improvements in flu tracking relative to history alone. This provides important lessons for query and social media-based flu tracking systems. Specifically, it demonstrates that forecasters need not use the conventional method of combining massive search query data to a relatively small number of CDC data points, and that creating a behavioral model of search is not only theoretically important but empirically effective.

## Methods
**Query labeling**. As we describe above, we use a hybrid human-computation method for labeling queries in the context of flu experience. We first define a labeling scheme for search queries. The scheme is designed to capture important differences in what the searcher is experiencing at the moment of executing a query, specifically flu-like experiences versus other experiences. We used simple flu-related key words to isolate a subset of queries and web pages from each respondent's web history to be labeled by trained coders. To ensure reliability of the coding, we used Kappa scores to assess intercoder reliability. Reliability on this set was Kappa = 0.865 for search queries and Kappa = 0.796 for web pages (see Supplementary Information 1.5.4).

Then, in order to generate an expansive list of flu-like queries for forecasting, we compute semantic distances between the positively labeled set of queries and other queries executed in the Bing search engine using the DOC2VEC method[17]. Based on this comparison, we retain all semantically related queries from Bing and label them according to our scheme. This allows us to identify additional flu-like queries for our time-series forecasting models. We also computed intercoder agreement and reliability scores for this set (see Supplementary Information 1.5.4).

**Lag selection and model evaluation**. For the SARIMA-X approach, we selected the appropriate orders ($p, P, q, Q$) for the model using AIC as a criterion. We set $p = 2$ in the Lasso-A1 model to account for seasonal effects in ILI rates. However, since we are using the Lasso penalty the coefficients of most of these lags will not appear in the final model. This approach also serves to select the appropriate lag in modeling ILI rates. Finally, we provide predictions for the 2015–2017 seasons by using a rolling 3-year period to train and predict.

We use a number of metrics to evaluate model performance. Each metric is some average of the deviation between the prediction and the true value. The first metric we report is root mean squared error (RMSE), which is defined as $\text{RMSE} = \sqrt{\frac{1}{n}\sum_{i=1}^{n}(y_i - \hat{y}_i)^2}$. This metric penalizes models more strongly when their predictions are far away from the true values. The second is mean average percent error (MAPE), which is defined as $\text{MAPE} = \frac{1}{n}\sum_{i=1}^{n}\left|\frac{y_i - \hat{y}_i}{y_i}\right|$. This metric is useful for understanding the average percent difference between the model's predicted values and the true values and so is fairly easy to interpret. The mean absolute error is similar to the MAPE except it does not take a percentage of the current value: $\text{MAE} = \frac{1}{n}\sum_{i=1}^{n}|y_i - \hat{y}_i|$.

**State flu data preprocessing and setup**. We collected flu data on the number of positive influenza swabs from the states of DE, DC, NM, and NY. To do so, we

downloaded each state's data in PDF form and put it into machine-readable format. These states were selected because they (1) collect and publicly post current flu prevalence rates on their official health pages and (2) make historical flu prevalence data available for long enough time spans to be useful for prediction. However, unlike the CDC data, these states do not provide a total number of hospital visits as the denominator to the ILI rate. Thus we only look at the raw counts of the ILI positive cases. Since we are predicting state-level ILI rates and search volumes can be relatively low, the exogenous search signals are often noisier than national-level signals.

Based on data availability we use a rolling 2–3 year period to train and predict our models. To compare the different methods we use RMSE as a metric after selecting the model parameters based on the AIC (Akaike Information Criterion). We tested the performance of our model on rolling 1- and 2-week ahead forecasts using exogenous signal at the most predictive lag.

**Reporting summary**. Further information on research design is available in the Nature Research Reporting Summary linked to this article.

## Data availability
The authors have deposited all replication materials, including minimal datasets required to replicate the methods used in this paper in a public GitHub repository located at: https://github.com/stefanjwojcik/ms_flu. All other relevant data are available upon reasonable request to the authors.

## Code availability
The authors have deposited all the necessary code for replicating the methods described in this paper in a public GitHub repository at: https://github.com/stefanjwojcik/ms_flu.

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

## Acknowledgements
This paper benefited from useful comments and insights from Dr. Devon Brewer and support from John Kahan. Special thanks to Thalita Coleman for Python programming support to extract data from PDF documents. IRB approval granted under Northeastern University # 18-07-03. This research is based upon work supported in part by the Office of the Director of National Intelligence (ODNI), Intelligence Advanced Research Projects Activity (IARPA), via 2017-17061500006. A.V. was partially supported by the NIGMS-NIH R01GM130668. The views and conclusions contained herein are those of the authors and should not be interpreted as necessarily representing the official policies, either expressed or implied, of ODNI, IARPA, or the US Government.

## Author contributions
A.B., J.F., R.J., G.K., D.L., and R.K. conceived of study. They also worked on the survey design, sampling regime, and fielded the survey. G.K. and D.L. offered substantial input on the statistical analysis and paper outline. S.W. analyzed the survey data, worked on MRP modeling, and led writing efforts. A.B. generated and tested time-series forecasting models. A.V. offered substantial input on statistical modeling and writing. All authors reviewed the document and offered feedback and ideas.

## Competing interests
The authors declare no competing interests.
