## [Peer Review File · Nature Communications]

Reviewers' comments:

Reviewer #1 (Remarks to the Author):

This is the second time I have reviewed the study by Wojcik et al and I see that while some of my previous concerns have been address, several of my main points have not. I will reiterate them here and still maintain their importance. In general I think the survey results are very interesting, novel, and an advance for the field and that the ILI prediction model detracts from the manuscript.

Major comments:

- The ILI prediction model is not necessary. First, from a philosophical point of view, this reviewer thinks there are a surfeit of ILI prediction algorithms – why do we need another? Second, the presented model does not outperform previous models – again, what does this add? Third, the claims that “The overshooting becomes more apparent for the history signal. MRP serves as to check this.” Is not correct – it seems random when the history is above MRP and vise-versa. I think including this weakens the paper.
- The language is better, but still needs work. I count the phrase “influenza-like-illness (ILI)” 7(!) times in the text. Please be consistent.
- The main text deals with A1 & A2 searches, while the supplement has A1, A2, B1, B2, C1, and D. Why are these other results not in the main text? These are the most interesting aspects of the paper.
- There needs to be more information on the survey in the ms. Where/who/how were the 20,000 selected? Who collected those data? How nationally representative is the study group? There isn't a citation for the data in the paper. Additionally: will these data be available to other researchers?
- The paper needs confidence intervals throughout. What's the magnitude of uncertainty for a relative risk comparing flu in house v. not of 1.57?
- Why is DE omitted from Figure 3?

Reviewer #2 (Remarks to the Author):

The goal of this study was to examine associations between symptoms and internet searches of influenza-like illness and to build a behavior-adjusted model to forecast ILI incidence with internet search terms. While other studies have examined population-level associations between ILI surveillance and internet search activity, this study provides novelty and interest in its validation of the relationship between disease and ILI-related search activity on an individual level and its attempt to adjust for non-representativeness in using internet search activity in a surveillance context.

While the study has collected valuable data in linking the use of digital data streams in surveillance and made progress in increasing the specificity of ILI-related searches, I think the authors could be more ambitious in their analyses and more applied in the framing and reporting of results in order to increase the value of their work in an epidemiological context. Alternatively, the authors may choose to target their work for a more statistical and machine-learning audience, in which case, the methods and comparisons of models should be strengthened and expanded upon. In the current manuscript, there seem to be critical descriptions of modeling methods that are missing, incorrect, or in need of clarification.

Primary comments:

- Considering the survey include a question on seeking care with a healthcare provider, I am surprised that the authors did not do any comparisons of this data and search behavior. Even outside of the development of forecasting models, one existing narrative about the advantage of digital epidemiology is its potential to identify disease outbreaks in “real-time” and among a

greater population, not just among those individuals that are captured in traditional surveillance systems several days later. It would be useful to gain insight into whether internet search behavior for ILI might precede or supplant visits to a healthcare provider, with the goal of describing the populations that might be captured in digital versus traditional surveillance.

- I am concerned about the period of symptom recall proffered in the survey question. Particularly when the ILI symptoms are mild, three months is a long recall period for an individual, much less for an individual to recall about other household members. There is literature to suggest that accurate recall is on the order of two days for an experience of diarrheal illness and up to two months if a hospital visit was required. I think it is particularly problematic that the directionality of recall bias could tend in both directions; on one hand, individuals may forget they had the symptoms since they are relatively common; alternatively, individuals may reason that there is a high probability they have experienced these symptoms in the past three months since they are relatively common. I suggest that the authors consider the implications for recall bias more seriously in their paper and identify ways in which this limitation may affect their results.

- What was the specific time range for the survey and did it correspond with active influenza activity? (This should be included in the main text.) With regards to my earlier concern about recall bias, are there differences between respondents that completed the survey closer to the start or peak of the influenza season?

- I think the Forecasting Methods section needs to include substantially more detail.

- o What was the volume, temporal scale, and geographic coverage of the Bing search queries from 2011 to 2016?

- o Where did the covariate demographic data (used in the MRP model) come from? At the very least, there needs to be a more clearly described section about these data in the supplement.

- o How were the forecasting models implemented (e.g., which software) and will code be shared?

- The MRP model described on page 4 needs to be further explained:

- o Can you please describe the response variable more clearly? I guess it is the proportion of search queries possessing an A1 search term in zipcode i ?

- o Perhaps I am missing something, but I do not understand how the first model equation is representing a smoothing and re-weighting process. The alphas appear to be regression coefficients, but is there also supposed to be some covariate data in the equation?

- o The equation indexes (i, j, k, p, q) in the model equations need to be defined. Presumably some of them represent bins, but it is not immediately clear what those bins are.

- o The second-to-last paragraph describes the application of a time series model over a rolling three-day window, but the primary model equation does not appear to have any time-varying components. I'm a bit confused in general about the time-varying nature of the MRP model; the demographic factors would vary across zipcodes, they will not vary much over time, so I don't understand the benefit of having this factor be time-varying.

- o Are the zipcode level models being run independently or jointly? Was there examination of spatial dependence? How is the MRP signal being aggregated to the national level?

- o Why is the prior for Income missing, and why is it described by beta when all other effects are described by alpha?

- o The indexes for the alpha equations do not match. For example, α_j^{State} is normally distributed for all h numbered 1 to 52? In any case, I would recommend the authors use different indexes so that there is less confusion with the p and q indexes in the SARIMA model.

- The authors should use consistent model names in the Forecasting Results, State Level Findings and Tables in the Results section. It would be good to introduce the model names when describing the model structures in the Methods as well.

- I recommend that the authors provide more descriptive captions for the tables and figures, and add the long-form terminology for metrics that are abbreviated.

- While it's important to show the model performance, I think the Results should report more results in the context of ILI rates. I think this will make the paper more relevant to an epidemiological audience. Are the models prone to under- or over-estimation at different times of the flu season? Are the models capable of capturing the peak timing and magnitude of seasonal outbreaks?

- I don't quite understand what is plotted in Figure 2B since the text describes these as 2-step ahead predictions. Which date does the x-axis represent? Were the plotted predictions were made two time-steps prior? Also, is there some reason why some of the models were excluded from the prediction figure? Regardless, it's curious that the model predictions seem to lag behind the actual ILI signal. As mentioned before, the authors should comment on the utility of these models in capturing peak timing and magnitude, not just with regards to model error.

- Limitations: Is it possible that multiple users are creating logged searches in the browser during the survey? Should the search term data represent more of a "household" measure instead of an individual measure?

Minor comments:

- Please add more references with greater specificity to sections in the supplement.

- Methods, Survey Data, third paragraph: The values reported do not sum to 654 survey respondents.

- Many sections of the supplement appear to be duplicates of the main text. This should be cleaned up and made less redundant.

- I don't think the supplement needs to include a section for the Discussion. All of the discussion points should be included in the main text.

Nature Comms Resubmission Memo

We thank the reviewers and editors for their thoughtful comments. We have sought to address every issue highlighted by the reviewers and editor. Consistent with the editor's note in response to our original submission, we did our best to address the underlying concerns and made changes based on the editor's instructions. We found the comments very helpful and sought to address them as fully as possible.

Responses to Reviewer 1

1. The ILI prediction model is not necessary. First, from a philosophical point of view, this reviewer thinks there are a surfeit of ILI prediction algorithms – why do we need another? Second, the presented model does not outperform previous models – again, what does this add? Third, the claims that “The overshooting becomes more apparent for the history signal. MRP serves as to check this.” Is not correct – it seems random when the history is above MRP and vice-versa. I think including this weakens the paper.
 - a. First, from a philosophical point of view, this reviewer thinks there are a surfeit of ILI prediction algorithms – why do we need another?
 - i. *Good point. We agree a new ILI model is not necessary, and we use an off the shelf model, and just add the signal derived. So, the scientific point here is not to produce the best possible ILI predictions-- that's not our value added-- it's to show if our signal helps in a standard model. We have adjusted our language to make this point clear. The issues with previous models have to do exactly with what we emphasize here - a different way of integrating search query data. We saw the need to establish the overall validity of using search queries as a proxy for flu-like experience. This is important for understanding how powerful search queries can be for predicting flu in the population. However, the counterargument from any potential reader would be say that our method may be valid in labeling queries that have a higher prior likelihood of being associated with the flu, but it may not be able to actually predict the flu. So, we sought to test whether this method would actually work to predict the flu.*
 - b. Second, the presented model does not outperform previous models – again, what does this add?
 - i. *On one hand, this paper is good news for papers that have used search queries to predict the flu - searches are associated with self-reported flu-like symptoms, thus proving the validity of the first assumption of existing methods. On the other hand, we think we have demonstrated a proof of concept of a survey-based method for forecasting the flu - one that does not require comparison against every search query in a particular engine - a method that prior work has been shown to be challenging for a variety of reasons. In general, we think the method of fitting such a large number*

of queries to a handful of data points needs revision. We think that such techniques are more likely than ours to fail without warning.

- c. Third, the claims that “The overshooting becomes more apparent for the history signal. MRP serves as to check this.” Is not correct – it seems random when the history is above MRP and vice-versa. I think including this weakens the paper.
 - i. Thanks. We removed the sentence on overshooting. We know that incorporating the MRP signal in the SARIMA-X model reduces the variance in the prediction error, so we emphasize that in the paper. The inclusion of the MRP as an exogenous signal has a considerable beneficial impact on the prediction quality.
2. The language is better, but still needs work. I count the phrase “influenza-like-illness (ILI)” 7(!) times in the text. Please be consistent.
 - a. *Many thanks for noting this issue - we have corrected the repetition of that phrase throughout. We have also reorganized and edited the paper for improved clarity.*
3. The main text deals with A1 & A2 searches, while the supplement has A1, A2, B1, B2, C1, and D. Why are these other results not in the main text? These are the most interesting aspects of the paper.
 - a. *We agree that the degree to which users search for illness information on the web is very interesting. We added a section to the main paper and another to the Supporting Information describing how often users reported searching for information online versus seeking information in-person from a healthcare provider (see SI:4). Based on input from the editor, we felt that adding each of the B1, B2, etc., category results to the main paper may distract from the main results of the paper. We kept them in the SI, however.*
4. There needs to be more information on the survey in the ms. Where/who/how were the 20,000 selected? Who collected those data? How nationally representative is the study group? There isn't a citation for the data in the paper. Additionally: will these data be available to other researchers?
 - a. *We report in section 3.1 of SI how the original panel is conducted. While we didn't have access to report the demographics of the full 20,000 panel, we do report the demographic characteristics of those we invited and those who responded to our survey (see SI 3.1 for these results).*
5. The paper needs confidence intervals throughout. What's the magnitude of uncertainty for a relative risk comparing flu in house v. not of 1.57?
 - a. *Many thanks for suggesting this. We have added these for our case-control model throughout. We used the 'zelig' package in R to generate simulation-based 95% confidence intervals for all the main quantities we report.*
6. Why is DE omitted from Figure 3?
 - a. *Fixed. Based on the feedback from the reviews, we moved figure 3 to the Supporting Information.*

Responses to Reviewer 2

7. Considering the survey include a question on seeking care with a healthcare provider, I am surprised that the authors did not do any comparisons of this data and search behavior. Even outside of the development of forecasting models, one existing narrative about the advantage of digital epidemiology is its potential to identify disease outbreaks in “real-time” and among a greater population, not just among those individuals that are captured in traditional surveillance systems several days later. It would be useful to gain insight into whether internet search behavior for ILI might precede or supplant visits to a healthcare provider, with the goal of describing the populations that might be captured in digital versus traditional surveillance.
- i. *We agree this is an interesting comparison to make and further motivates the core findings in the paper. We added analysis to the main document and the supporting information to address it. To incorporate the idea behind this comment, we looked at differences in search rates between those who sought information about symptoms from a healthcare provider versus those who sought information from the Internet. We found that only 6% of respondents saying they asked for information from their healthcare provider and did not look online. By contrast, 33% reported looking for information online and did not ask their provider, and 26% reported looking online **and** asking their healthcare provider. We now dedicate space to this in the survey findings section of the main paper, and address this question in the supporting information (see section “Information from Healthcare Providers vs. the Internet”).*
 - b. I am concerned about the period of symptom recall proffered in the survey question. Particularly when the ILI symptoms are mild, three months is a long recall period for an individual, much less for an individual to recall about other household members. There is literature to suggest that accurate recall is on the order of two days for an experience of diarrheal illness and up to two months if a hospital visit was required. I think it is particularly problematic that the directionality of recall bias could tend in both directions; on one hand, individuals may forget they had the symptoms since they are relatively common; alternatively, individuals may reason that there is a high probability they have experienced these symptoms in the past three months since they are relatively common. I suggest that the authors consider the implications for recall bias more seriously in their paper and identify ways in which this limitation may affect their results.
 - i. *We have added citations to literature that tackles issues of recall. We now go into greater detail to describe the problems of recall relating to the length of time between the event and the survey in the limitations section of the main paper and in the supporting information. Specifically, we cite sources showing that memory is more effective for recent events, which will create a recency effect. Dating error increases in a linear fashion as period increases - though remains unbiased. These are located in the limitations section in the main paper and in SI: 3.4.*

- c. What was the specific time range for the survey and did it correspond with active influenza activity? (This should be included in the main text.) With regards to my earlier concern about recall bias, are there differences between respondents that completed the survey closer to the start or peak of the influenza season?
- i. *We fielded an initial survey wave from March 19th to March 27th 2015, then a second wave from April 27th to April 31. The flu season typically peaks in February, but sometimes peaks in December, January, and March. During the 2014-2015 season, the flu peaked in December (<https://www.cdc.gov/flu/pastseasons/1415season.htm>). Browser data was collected between Nov. 1, 2014 and continued until March 31, 2015. We added a sentence on this to the survey methods section of the main text and go into detail in the SI (SI: 3.2).*
 - ii. *We checked whether respondents who completed the survey closer to the start of the season were more or less likely to report flu symptoms personally or in the household. We looked at differences between these two survey waves. We found no statistically significant differences in the rates of flu symptom reporting between the early and late responders. Including whether the respondent submitted their survey in the earlier cohort did not substantively alter the coefficient on the household flu variable in our main model either. We have added discussions and tables discussing this to the SI (Si: 3.4).*
8. I think the Forecasting Methods section needs to include substantially more detail. What was the volume, temporal scale, and geographic coverage of the Bing search queries from 2011 to 2016? Where did the covariate demographic data (used in the MRP model) come from? At the very least, there needs to be a more clearly described section about these data in the supplement. How were the forecasting models implemented (e.g., which software) and will code be shared?
- a. What was the volume, temporal scale, and geographic coverage of the Bing search queries from 2011 to 2016?
 - i. *We are not permitted to share the total numbers of searches for any given query from Bing. Instead, we generated z-scores for three core quantities across all states and years: the total number of searches, the total number of searches related to flu (again drawing on the DOC2VEC expansion technique), and the total number of A1 searches (based on human labels). This is similar to 'Google Trends' data. For the state-level estimates, we first divided the number of queries by the state's population to create a per capita per year estimate. We then averaged this estimate by year to create an annual state estimate. Finally, we subtracted the overall annual state average and divided by the standard deviation to create a z-score.*
 - b. Where did the covariate demographic data (used in the MRP model) come from? At the very least, there needs to be a more clearly described section about these data in the supplement.

- i. *We have added more information about this in the main document and in the SI (SI: 6.6). The variables education, age, and the number of children per house came from the American Community Survey (2014 and 2015, 5 year estimates). They were downloaded via the American Community Survey Application Programming Interface (API) using the ACS package in R (<https://cran.r-project.org/web/packages/acs/acs.pdf>). Education is a binned (by quantile) measure of the proportion of individuals in each zip code who had completed a post-high school degree program, divided by the population of the zip code (these came from the 2014 5-year ACS estimates). Age is a binned (by quantile) measure of the median age in a zip code (from 2014 5-year ACS estimates). Finally, the number of children per house is a binned quantile measure of the average number of children in each household within a zip code (also from the 2014 5-year ACS estimates).*
- c. How were the forecasting models implemented (e.g., which software) and will code be shared?
 - i. *The MRP models were implemented in R using the lme4 package (Bates et al. 2015). The forecasting models were implemented in R using the Forecast package by Rob Hyndman. Code will be made available as supplementary files.*

9. Fixes to MRP description:

- a. Can you please describe the response variable more clearly? I guess it is the proportion of search queries possessing an A1 search term in zipcode i ?
 - i. *That interpretation is correct for the MRP section - the proportion of search queries possessing an A1 search term in zipcode i . We edited the main text and supplementary file to make the response variable in the MRP section more clear. The window time component was not represented in the main text, following a similar representation as Wang et al. (2015), but we added it for additional clarity. Additionally, we have otherwise edited the equations for consistency in indexing and readability.*
- b. Perhaps I am missing something, but I do not understand how the first model equation is representing a smoothing and re-weighting process. The alphas appear to be regression coefficients, but is there also supposed to be some covariate data in the equation?
 - i. *We have made edits to clarify this in the main document and in the supporting information. As written in the original submission, we did not show the formula for re-weighting process, but we have added this to the main text for full clarity. We have updated the model equations to make the terms clearer in addition to the equation for poststratification. The model is based on Multilevel Regression and Poststratification (so-called MRP) models described elsewhere (see e.g Wei, et al., 2014, Lax and Phillips, 2009). We emphasize in the text that the idea behind the smoothing is that zip codes with distinct combinations of demographic features will be rare, and thus suffer from variability due to small sample*

- sizes. By creating priors for demographic variables, the multilevel model borrows some power from zip codes with similar characteristics. The poststratification is conducted by generating weights based on the true prevalence of zipcodes of type i , then weights the data accordingly.*
- c. The equation indexes (i, j, k, p, q) in the model equations need to be defined. Presumably some of them represent bins, but it is not immediately clear what those bins are.
 - i. Yes, a majority of the variables are binned quantiles, with the exception of income which is a fixed coefficient (following advice from Buttice and Highton (2013), see more explanation below). We have added further elaboration to the model equations to more clearly depict what each index means and added explicit statement in the text about what is being binned and what is not.*
 - d. The second-to-last paragraph describes the application of a time series model over a rolling three-day window, but the primary model equation does not appear to have any time-varying components.
 - i. We appreciate this point. We sought to follow the clearest possible representation of the model from the literature for consistency and readability. We have added subscripts (t) to indicate the time window and hopefully clarify the model for all readers. The t subscript now depicts the time window in which the model is being estimated.*
 - e. I'm a bit confused in general about the time-varying nature of the MRP model; the demographic factors would vary across zipcodes, they will not vary much over time, so I don't understand the benefit of having this factor be time-varying.
 - i. Correct, the demographic variables do not change over time, but the underlying flu rate within each zip code does change over time, and the MRP model smooths and reweights the average prevalence of flu-like searches to capture this change. We have tried to make this clearer in the present draft. Our paper follows a similar structure as Wang et al. (2015), in which they apply a MRP model over a moving window to survey data from a video game console. Similarly, a moving window allows us to generate a smoothed and reweighted average for the last day of the time window in every zip code or geographic area desired. We then use the smoothed signal from this method as input to our time series model.*
 - f. Are the zipcode level models being run independently or jointly? Was there examination of spatial dependence? How is the MRP signal being aggregated to the national level?
 - i. Models are estimated with a multilevel model (so-called 'partial pooling'), and therefore are not fully disaggregated by zipcode nor fully pooled. Each row in the query data represents one query at a period of time in a particular zipcode and state. We have varying densities of searches in different zip codes, which makes it very challenging to fully disaggregate by zip code. Instead, the multilevel model will treat zip codes with greater search densities as stronger signal than zip codes with lesser search*

densities, and borrow statistical power from demographically similar zip codes. Regarding spatial correlation, the state where the zip code is located is the only spatial variable in the model. Spatial dependence will make the standard errors of the model inaccurate, but not the coefficient estimates. We do not use the MRP model for calculating statistical significance measures - the MRP technique is only used as a smoothing and re-weighting step of the flu signal. We tried to clarify these points in the paper.

- g. Why is the prior for Income missing, and why is it described by beta when all other effects are described by alpha?
 - i. Here, we follow the advice of Buttice and Highton (2013) and the example of other MRP papers to include one state-level covariate in our models. In their work, they show that including at least one state-level covariate increases the correlation and reduces absolute bias between the final signal and the underlying true value. We now include this point in the main text in the MRP methods section.*
 - h. The indexes for the alpha equations do not match. For example, α_j^{State} is normally distributed for all h numbered 1 to 52? In any case, I would recommend the authors use different indexes so that there is less confusion with the p and q indexes in the SARIMA model.
 - i. We appreciate this point. There was a typo in the section of equations displaying distributional assumptions, which we have now fixed. The assumption we were trying to convey (and we think we now convey better!) was that the alphas for the states are normally distributed, which is an assumption we made for all the terms described by alphas. We now recognize that the distributional assumptions were not particularly helpful to a general audience so we moved them to the supporting information and corrected the mismatched alphas.*
 - i. The authors should use consistent model names in the Forecasting Results, State Level Findings and Tables in the Results section. It would be good to introduce the model names when describing the model structures in the Methods as well.
 - i. We made edits to standardize mentions of the 'behavioral' model and the 'tracking' model throughout the main text. We also added text to the introduction of the methods section to introduce definitions for these models.*
10. I recommend that the authors provide more descriptive captions for the tables and figures, and add the long-form terminology for metrics that are abbreviated.
- a. We have updated the captions on the tables in the main text, and edited text to avoid short-form abbreviations where it could be confusing.*
11. While it's important to show the model performance, I think the Results should report more results in the context of ILI rates. I think this will make the paper more relevant to an epidemiological audience. Are the models prone to under- or over-estimation at

different times of the flu season? Are the models capable of capturing the peak timing and magnitude of seasonal outbreaks?

- a. While it's important to show the model performance, I think the Results should report more results in the context of ILI rates. I think this will make the paper more relevant to an epidemiological audience.
 - i. *We have edited to deliver a more substantive interpretation of the results in the context of ILI in the main text. Specifically, we now discuss the raw correlation between the MRP input signal and the flu rates (~90). We also describe in terms of the overall rate of flu, the percentage away we were from a perfect prediction.*
 - b. Are the models prone to under- or over-estimation at different times of the flu season? Are the models capable of capturing the peak timing and magnitude of seasonal outbreaks? The authors should comment on the utility of these models in capturing peak timing and magnitude, not just with regards to model error.
 - i. *For the national model, we found that models based on history were more likely to overestimate the peaks relative to the MRP-based model. We comment on this in the figure in the forecasting results section. Based on this and other comments throughout, we include more substantive interpretation of these results in the main results section.*
12. I don't quite understand what is plotted in Figure 2B since the text describes these as 2-step ahead predictions. Which date does the x-axis represent? Were the plotted predictions made two time-steps prior? Also, is there some reason why some of the models were excluded from the prediction figure? Regardless, it's curious that the model predictions seem to lag behind the actual ILI signal. As mentioned before, the authors should comment on the utility of these models in capturing peak timing and magnitude, not just with regards to model error.
- a. *I don't quite understand what is plotted in Figure 2B since the text describes these as 2-step ahead predictions. Which date does the x-axis represent? Were the plotted predictions made two time-steps prior?*
 - i. *The x-axis corresponds to the current date. Correct, the prediction lines at a given date correspond to the two-step ahead prediction for that day. We have clarified.*
 - b. Also, is there some reason why some of the models were excluded from the prediction figure? Regardless, it's curious that the model predictions seem to lag behind the actual ILI signal. As mentioned before, the authors should comment on the utility of these models in capturing peak timing and magnitude, not just with regards to model error.
 - i. *Delaware data is more sparse compared to the other states, with low counts and as such susceptible to more erratic and unreliable predictions (for all methods, not just ours), so we initially omitted it from the figure, even though the MRP model still displays lower prediction error compared to the historical models. We now provide all the absolute error plots in the Supporting Information. Regarding lagging of the prediction signal from the ILI signal, it is somewhat hard to dissect the trajectory of the*

predictions, in that undershooting at one time point will propagate to the future. The model also incorporates a good amount of information from prior flu levels (the underlying history, which is extremely powerful), so the MRP model prediction will partially reflect the prior values in its prediction. Regarding peak timing and magnitude, our model optimizes on MSE so it is challenging to meaningfully interpret these results in light of other metrics that occur only once per season. We try to make clear in the paper that the model is meant to demonstrate that the MRP survey signal provides a utility over strong historical baselines, rather than establishing a new state-of-the-art.

13. Limitations: Is it possible that multiple users are creating logged searches in the browser during the survey? Should the search term data represent more of a “household” measure instead of an individual measure?

a. This is a good point, and indeed the household measure is plausible. To check for this, when we restrict the data to cases where the respondent indicates that she is the primary user of the machine where the browsing tracker is installed, the results are similar. We include a discussion of the challenges of associating searches from a particular machine with a single user.

14. Minor comments:

a. Please add more references with greater specificity to sections in the supplement.

i. Done, thank you for noting it.

b. Methods, Survey Data, third paragraph: The values reported do not sum to 654 survey respondents.

i. The values in this section report omit 10 respondents who did not have any search volumes. We added a note to the document to clarify.

c. Many sections of the supplement appear to be duplicates of the main text. This should be cleaned up and made less redundant.

i. Thank you, we have removed these redundancies.

d. I don't think the supplement needs to include a section for the Discussion. All of the discussion points should be included in the main text.

i. Noted, we have removed the discussion from the supplement and placed all critical points in the main text.

Reviewers' comments:

Reviewer #1 (Remarks to the Author):

The authors have addressed all of my concerns.

Reviewer #2 (Remarks to the Author):

The methods and model descriptions are much improved in this revision and the flow of the paper is much easier to follow. However, I do not find that the new sections/text responding to my comments on 1) improving the relevance of the results to an epidemiological audience and 2) addressing the limitations of the data and survey questions add as much value or context to the revised manuscript as I would have hoped. In response to the other reviewer, the authors note that they have shifted the value-added of their manuscript – focusing on whether MRP improves upon previous models rather than making excellent ILI predictions. I think that this shift in focus is somewhat incompatible with my suggestions on how to expand the impact of the work to a wider audience. I feel that the comment from my previous review still stands:

“I think the authors could be more ambitious in their analyses and more applied in the framing and reporting of results in order to increase the value of their work in an epidemiological context. Alternatively, the authors may choose to target their work for a more statistical and machine-learning audience, in which case, the methods and comparisons of models should be strengthened and expanded upon.”

Given the shift in focus and the relatively poor absolute performance of the model predictions (based on visual comparison in Figure 2b), I suggest that the authors consider this second option more carefully, as I think that the MRP method is interesting and worth additional consideration in future implementations of these predictive models. If the authors do decide to go this route, I think it will be important to emphasize whether the base model + MRP outperforms predictions using the base model + other exogenous forcing term and to discuss the attributes that may contribute to its improved performance.

Other comments:

The results for Figure 2b should include confidence intervals and be discussed and cited in the main text. What are the units for the Y-axis? I guess it is Prediction of ILI rate? Also, I'm not sure about the author's interpretation of these results in saying that history-based predictions often over-estimate the peaks. Aside from the predictions not being well-aligned with the actual signal, both the SARIMA-HIST and SARIMA-MRP models seem to over-estimate peaks and under-estimate troughs.

The Figure 2 and Table 3 captions should describe the data presented in the Figure and Table (not the interpretation of the results).

Tracking Results Para. 1: The correlation coefficient needs to be reported with the specific test and hypothesis. Also, I understand neither how to interpret “0.90/0.84” and “12.29/13.74%” nor what the percentage error represents (e.g., how was it calculated?, percentage with respect to what?). Maybe the Methods section should include a subsection on Model Evaluation so that these and other model diagnostics can be described explicitly.

Pg. 4 has two notations for the income coefficient ($\beta_{[it]}^{\text{Income}}$ and $\beta_{1[i][t]}^{\text{Income}}$). I think this is a typo.

Elizabeth C. Lee

Reviewer #2 (Remarks to the Author):

The methods and model descriptions are much improved in this revision and the flow of the paper is much easier to follow. However, I do not find that the new sections/text responding to my comments on 1) improving the relevance of the results to an epidemiological audience and 2) addressing the limitations of the data and survey questions add as much value or context to the revised manuscript as I would have hoped. In response to the other reviewer, the authors note that they have shifted the value-added of their manuscript – focusing on whether MRP improves upon previous models rather than making excellent ILI predictions. I think that this shift in focus is somewhat incompatible with my suggestions on how to expand the impact of the work to a wider audience. I feel that the comment from my previous review still stands:

“I think the authors could be more ambitious in their analyses and more applied in the framing and reporting of results in order to increase the value of their work in an epidemiological context. Alternatively, the authors may choose to target their work for a more statistical and machine-learning audience, in which case, the methods and comparisons of models should be strengthened and expanded upon.”

Given the shift in focus and the relatively poor absolute performance of the model predictions (based on visual comparison in Figure 2b), I suggest that the authors consider this second option more carefully, as I think that the MRP method is interesting and worth additional consideration in future implementations of these predictive models. If the authors do decide to go this route, I think it will be important to emphasize whether the base model + MRP outperforms predictions using the base model + other exogenous forcing term and to discuss the attributes that may contribute to its improved performance.

- *Thanks. We certainly discussed the challenges of meeting the expectations of both reviewers given the tension between some of the suggested revisions. We agree that the value of the paper may speak more to a machine learning and statistical audience as compared to an epidemiological audience, although we think the epidemiological community is important to consider and we hope it will find value in our method. In order to strengthen the paper toward a machine learning/statistics audience, we have included several additional details about how our model was built, where we see our model fitting in the literature, and why such an approach tends to work. We specify all these changes in detail below:*

1) We have edited the discussion of the MRP to explain in more detail why it is useful for practitioners from many disciplines. Specifically, we note in the beginning of the MRP section that the MRP allows for modeling a wider array of data than models that require nationally-representative data. The MRP is effective at smoothing and re-weighting to known population benchmarks, and

we note that many existing research datasets are not collected using probability sampling so may benefit from this method.

2) We added more information about how the MRP model helps reduce noise for datasets where coverage for population subgroups is sparse (MRP section paragraphs 1 and 3). We think readers will appreciate the fact that the MRP can make adjustments to compensate for sparse sampling among critical subgroups who may be challenging to include in some studies. For example, if poor and minority groups are sparsely sampled, the MRP can reduce the variance estimates among these groups by borrowing statistical power from groups that are similar.

3) We now further emphasize in the paper that this method could be useful in tracking various other phenomena for which survey data might provide an empirical link between large-scale digital sources such as queries or tweets and regional outcomes such as ILI, unemployment rates, movie ticket sales, or other aggregated outcomes (see main paper page 2 paragraph 1).

4) We have added a discussion of the comparison between the processed MRP model and one with an unprocessed exogenous forcing term (the raw A1 term). We agree it is important for readers of this paper to understand comparisons between the final MRP models and relevant comparisons such as historical models and those with alternative exogenous forcing terms. We now do more in the paper to contextualize the performance of our model and make relevant comparisons to existing models (see p2 para 2). We would love to compare our model against zipcode level queries from another search engine, but these data are simply impossible to access. We also now note in the paper that while we frame it as a proof-of-concept, we do think our model shows impressive accuracy - as it beats an autoregressive LASSO model with seasonal effects whose lags are chosen based on out-of-sample performance (we note this on page 2 paragraph 2). It is also worth pointing out that many models do not compare against such challenging historical baselines. For instance, the ARGO model (Yang et al 2015) uses LASSO to select the appropriate number of autoregressive lags for its main model, but it does not do the same for its historical comparison model - the authors use an unprocessed prior CDC value to represent a historical model and models containing Google trends and up to three lag terms.

5) Regarding the issues of weaknesses of surveys, we assume the reviewer is referring to recall issues. We have added more citations to revisit the issue of survey recall, but this time the citations focus on recall in clinical settings. Although slightly different results surface in the clinical literature, a number of highly cited studies find underreporting when the recall period is extended and overreporting when the recall period is reduced. This stands somewhat in

contrast to the public opinion survey literature, which appears more oriented toward the idea that recall becomes more highly varied as the recall period extends, but remains unbiased overall. We now note in the paper that Boerma (1991) find under-reporting of diarrhoea if the recall period is longer than 2–3 days, but over-reporting of very recent or current diarrhoea. The study also finds that reporting errors tend to vary between countries, so it is somewhat challenging to generalize these results to a US context. Perhaps more relevant to our current study, Arnold (2013) show that the ideal period of recall for diarrhea, fever, and cough that reduces both variance and bias is about seven days for caregivers based on data from India. This is based on the fact that reducing the window reduces the amount of data by greater than 40% for certain outcomes. Another relevant study due to the focus on caregivers is Overbey (2019), who show that when comparing recall of diarrheal illness in children by caregivers, a one-week recall period shows substantially greater rates of diarrhea prevalence compared to a two-week recall period. This is in line with the earlier comment that extending the recall period tends to lead to underreporting. The consequences for our study are that if recall leads to underreporting, then the models are biased toward zero. In our case that would mean the coefficient for the relative rate of flu search is likely higher than what we observe, and our model could be considered conservative in this respect.

Other comments:

{REGARDING FIGURE 2B}

The results for Figure 2b should include confidence intervals and be discussed and cited in the main text.

- *We have added confidence intervals for both the $h=1$ and 2-step ahead errors, where the 2-step ahead prediction corresponds to the prediction 2-weeks in the future using the concurrent search signal. This 2-step ahead prediction estimate is expected to have a higher uncertainty but is useful since the CDC estimate can take up to 2 weeks to update.*

What are the units for the Y-axis? I guess it is Prediction of ILI rate?

- *Yes, this is the national percentage of ILI visits according to the CDC. We have updated the paper to make this clearer.*

Also, I'm not sure about the author's interpretation of these results in saying that history-based predictions often over-estimate the peaks. Aside from the predictions not being well-aligned with the actual signal, both the SARIMA-HIST and SARIMA-MRP models seem to over-estimate peaks and under-estimate troughs.

- *In the previous draft the figure 2b) corresponded to a 2-step ahead prediction and so had a larger error compared to the observed CDC estimate (which is expected since this forecast used a 2-week old observed value for prediction). For comparison we have*

included a 1-step ahead prediction with better alignment with the observed signal and tighter CIs.

- *We have changed the caption in the figure in line with the other comment so that it is focused completely on what the figure is depicting. As a consequence, this interpretation has been struck.*

{FIGURES 2 and 3}

The Figure 2 and Table 3 captions should describe the data presented in the Figure and Table (not the interpretation of the results).

- *We have added a description of the data in the figure here and moved all interpretation to the main text.*

{TRACKING RESULTS P1}

Tracking Results Para. 1: The correlation coefficient needs to be reported with the specific test and hypothesis. Also, I understand neither how to interpret “0.90/0.84” and “12.29/13.74%” nor what the percentage error represents (e.g., how was it calculated?, percentage with respect to what?). Maybe the Methods section should include a subsection on Model Evaluation so that these and other model diagnostics can be described explicitly.

- *When reporting the correlation coefficients we thought it would add context to let the reader know the raw correlation between the model prediction and the outcome, but perhaps it added more confusion. We have added detail to this section to describe how to interpret these quantities. We now note in the paper that the “12.29/13.74%” refers to the percent error of the forecast to the true value in 2016/2017, respectively.*
- *At R2’s request, we have also added a section to the Supporting Information with explicit mathematical definitions for the metrics we report. We also describe how to interpret these metrics in the context of model evaluation. See SI: 6.8.*

{P4 COEFFICIENTS}

Pg. 4 has two notations for the income coefficient ($\beta_{[it]}^{\text{Income}}$ and $\beta_{1[i][t]}^{\text{Income}}$). I think this is a typo.

- *Good catch. Fixed, thank you!*

REVIEWERS' COMMENTS:

Reviewer #2 (Remarks to the Author):

My sincere apologies for the long delay. I am satisfied with the author's edits to the paper and in the end I think the paper strikes a nice balance for a wide variety of audiences. More than ever we need new approaches to incorporate human behavior and under-sampled populations into models of infectious disease dynamics and these results represent a forward step in that direction.

Elizabeth C. Lee